# The Role of SCL Isoforms in Embryonic Hematopoiesis

**DOI:** 10.3390/ijms24076427

**Published:** 2023-03-29

**Authors:** Chin-Kai Chuang, Su-Fen Chen, Yu-Hsiu Su, Wei-Hsin Chen, Wei-Ming Lin, I-Ching Wang, Song-Kun Shyue

**Affiliations:** 1Division of Animal Technology, Animal Technology Research Center, Agricultural Technology Research Institute, No.52, Kedong 2nd Rd., Zhunan Township, Miaoli County 35053, Taiwan; 2Institute of Biomedical Sciences, Academia Sinica, 128 Sec. 2, Academia Rd. Nankang, Taipei 115, Taiwan; 3Institute of Biotechnology, National Tsing Hua University, No. 101, Sec. 2, Kuang-Fu Road, Hsinchu 30013, Taiwan; 4Department of Life Science, National Tsing Hua University, No. 101, Sec. 2, Kuang-Fu Road, Hsinchu 30013, Taiwan; 5Brain Research Center, National Tsing Hua University, No. 101, Sec. 2, Kuang-Fu Road, Hsinchu 30013, Taiwan

**Keywords:** SCL, Etv2, primitive hematopoiesis, definitive hematopoiesis, gene editing

## Abstract

Three waves of hematopoiesis occur in the mouse embryo. The primitive hematopoiesis appears as blood islands in the extra embryonic yolk sac at E7.5. The extra embryonic pro-definitive hematopoiesis launches in late E8 and the embryonic definitive one turns on at E10.5 indicated by the emergence of hemogenic endothelial cells on the inner wall of the extra embryonic arteries and the embryonic aorta. To study the roles of SCL protein isoforms in murine hematopoiesis, the SCL-large (SCL-L) isoform was selectively destroyed with the remaining SCL-small (SCL-S) isoform intact. It was demonstrated that SCL-S was specifically expressed in the hemogenic endothelial cells (HECs) and SCL-L was only detected in the dispersed cells after budding from HECs. The *SCL*^Δ/Δ^ homozygous mutant embryos only survived to E10.5 with normal extra embryonic vessels and red blood cells. In wild-type mouse embryos, a layer of neatly aligned CD34^+^ and CD43^+^ cells appeared on the endothelial wall of the aorta of the E10.5 fetus. However, the cells at the same site expressed CD31 rather than CD34 and/or CD43 in the E10.5 *SCL*^Δ/Δ^ embryo, indicating that only the endothelial lineage was developed. These results reveal that the SCL-S is sufficient to sustain the primitive hematopoiesis and SCL-L is necessary to launch the definitive hematopoiesis.

## 1. Introduction

Three successive waves of transient hematopoiesis coordinate to establish the murine embryonic blood system. The primitive hematopoiesis is initiated at embryonic day 7.5 (E7.5) from the cells inside blood islands in the extra embryonic yolk sac [1,2]. As the number of blood islands increases, the outer layer of the accumulated islands fuse with each other to form a network called the capillary plexus at the four-somite stage (E8.25) [3,4], and this plexus is connected with the embryonic circulatory system after the six-somite stage (E8.5) [3] through the vitelline vessel accompanied by the initiation of the heart beat [5]. Then, the plexus is remodeled into the blood vessel system by hemodynamic force [6]. A different type of hematopoiesis, the definitive hematopoiesis, is derived from the hemogenic endothelial cells (HECs) on the walls of certain arteries. The extra embryonic pro-definitive hematopoiesis launches at around the 12- to 19-somite stage (E9.0) at which point slow plasma flow is detected between the extra and intra blood vessel networks [6]. The intra embryonic definitive hematopoiesis emerges at E10.5, indicated by the phenomenon of endothelial to hematopoietic transition (EHT) which was initially, even though rarely, observed in aorta where hematopoietic stem cells (HSCs) were detected budding from the CD34^+^ endothelial cells (termed the hemogenic endothelial cell, HEC) layer [7,8]. The emergence of HSCs from aorta HECs becomes frequent from E11.5 to E12.5 [9,10,11] and these HSCs migrate, colonize, and expand in fetal liver where the number of HSCs increases up to 38 fold from E12 to E16 [12]. Then, HSCs begin to shift to the bone marrow to establish adult hematopoiesis at E17 [13].

Since the *Etv2* (originally named *Etsrp71*) gene is essential for the development of blood islands, no blood cells and vessels can be found in *Etv2* null mouse embryos [14]. No blood cells are found in *SCL* (also known as *Tal-1*) null mouse embryos either, whilst the blood vessel system is developed, suggesting that SCL is necessary for embryonic hematogenesis [15,16]. Two differentially expressed isoforms of *SCL* mRNA were detected temporally and spatially in zebrafish embryo. A short *SCL* mRNA isoform (Scl-β) was expressed earlier at the 1- to 2-somite stage, peaking at around the 18-somite stage, and decreasing dramatically after 2 days post fertilization (2 dpf) in a pair of lateral stripes, in the posterior lateral mesoderm (PLM). On the other hand, the long *SCL* mRNA isoform (Scl-α) was expressed at the 5- to 6-somite stage, peaking at around the 18-somite stage, and remaining at a high expression level at 2 dpf, and is dominant in adult kidney where the major hematopoiesis takes place [17]. The Scl-α and Scl-β mRNA are derived from a distal and proximal promoter, respectively, and the latter was more sensitively regulated by Etv2. In addition, the expression of Scl-β was required for the specification of hemogenic endothelium and Scl-α was necessary for the maintenance of HSCs in the aorta of zebrafish [18,19]. Although the development of embryonic hematopoiesis is quite conserved in vertebrates [20], only limited clues about mammalian SCL isoforms have been found hitherto. In comparison with the fish data, two different transcriptional initiation sites at exon Ia and Ib of the *SCL* gene have been reported in the genome of the mouse F4N myeloid leukemia cell line. The promoter for the upstream exon Ia was tightly activated by GATA1 derived transcripts encoding a full length 329 amino acid SCL protein (SCL-L) translated from the initiation codon in exon IV, whereas the promoter for the downstream exon Ib was GATA1 independent and transcribed a short transcript without exon IV (Appendix A). The last *SCL* mRNA isoform corresponding to the fish Scl-β isoform encoded the small form of the SCL protein (SCL-S) which is the C-terminal 154 amino acid residues of SCL-L [21,22,23].

Different roles might be performed by SCL-L and SCL-S in murine primitive and definitive hematopoiesis. Taking advantage of the precise gene editing ability of the CRISPR/Cas9 system [24,25], the mouse SCL-L mRNA comprising exon IV can be selectively destroyed while SCL-S mRNA remains intact. We examined the roles of both SCL isoforms in hematopoiesis and showed that deletion of the SCL-L hindered the definitive hematopoiesis but not the primitive hematopoiesis in embryonic development.

## 2. Results

### 2.1. Preparation of the Gene Edited Mice

In contrast to the conventional gene knockout method in which an antibiotic resistant gene cassette was usually used to interrupt a target gene by homologous recombination, the gene editing by the CRISPR/Cas9 system is mediated by the cut and nonhomologous-end-joining mechanism [26]. It is efficient for eradicating a DNA fragment from the genome simply by using a pair of single-guide RNAs and Cas9 [24,27]. According to the same strategy, a 118 bp DNA fragment in exon IV of the mouse *SCL* gene was deleted to create an out of frame mutation (Figure 1A). Because of the deletion, genotyping could be simply performed by agarose gel electrophoresis (Figure 1B). The heterozygote of the deletion mutants (*SCL*^+/Δ^) grew normally like the wild type; however, the homozygous fetuses (*SCL*^Δ/Δ^) could be found by E11.5. The heritable probability of this deletion mutation followed Mendel’s laws up to E10.5.

### 2.2. Characterization of the Gene Edited Murine Fetuses

Because there were no homozygous *SCL*^Δ/Δ^ fetuses that could be found at E11.5, E10.5 fetuses were collected to study the developmental status of hematopoiesis under a stereomicroscope. It was intriguing that besides the wild type and heterozygous mutant, red blood cells were clearly present in the yolk sacs (Figure 1C) and embryo proper (Figure 1D) of the homozygous mutants. The reason why the homozygous mutants could survive by E11.5 was supposed to be due to some defects in the second pro-definitive and/or the third definitive waves of embryonic hematopoiesis. The distribution of the CD34^+^ HECs in both yolk sac and embryo proper at E10 was localized by whole mount immunofluorescence staining. A lateral view taken by confocal microscope is shown in Figure 2A. Strong clustered and spotted CD34 signals were revealed in the extra embryonic tissue as well as a thin but obvious line of CD34 signal indicating the ventral side of the aorta inside the embryo proper. A higher magnification image was taken to unveil two types, clustered-rectangular and dispersed-round, CD34^+^ cells that were present in CD31 defined tubular structures in the yolk sac (Figure 2B).

### 2.3. Expression Pattern of SCL Isoforms

The embryonic definitive hematopoiesis was initiated at E10.5 [7] and live homozygous *SCL*^Δ/Δ^ fetuses could be isolated up to E10.5, therefore, E9.5 and E10.5 were set as analysis check points for the extra embryonic pro-definitive hematopoiesis and intra embryonic definitive hematopoiesis, respectively. Two antibodies, a mouse monoclonal antibody against the C-terminal part of SCL (SCL-C) and a rabbit polyclonal antibody against the N-terminal part of SCL (SCL-N), were selected to distinguish the expression patterns of SCL-L and SCL-S isoforms. The expression of SCL-L was indicated by overlapped staining of both SCL-N and SCL-C, nevertheless SCL-S could be only recognized by SCL-C. Clusters of partially aligned SCL-C positive cells were observed in the E9.5 wild-type yolk sac and a fraction of these SCL-C positive cells were stained by SCL-N as well. On the other hand, much stronger SCL-C signals alone were detected in the *SCL*^Δ/Δ^ yolk sac (Figure 3A). Embryonic sections across the aorta (Appendix A) were manipulated to illustrate the state of hematopoiesis inside the embryo. SCL-S was restrictively located in the endothelial layer of the aorta of both wild type and *SCL*^Δ/Δ^ mutants at E9.5 (Figure 3B) as well as at E10.5 (Figure 3C). Meanwhile SCL-L was just detected in dispersed cells inside the cavities of wild-type trunks.

### 2.4. Development of Aortic HECs

The definitive hematopoietic stem cells were generated from HECs [7,8,28,29]. In comparison with the other types of vascular endothelial cells, CD34 is a quite specific marker for hemogenic endothelium and CD31 is a general marker for all [30]. In vitro differentiation experiments concluded that CD43 defined definitive hematopoietic stem/progenitor cells from human embryonic stem cells [31,32]. Unlike the SCL-S, all the CD31, CD34, and CD43 markers were still not observed on the endothelial layer of E9.5 aorta (Figure 4A), instead, dispersed CD31^+^/CD34^+^/CD43^+^ cells were found in the vessel cavities (Figure 4A,B) and weak but significant c-Kit and CD45 markers were revealed on most of these cells (Figure 4C). Such data suggested that the dispersed cells were hematopoietic progenitor/stem cells. At E10.5, significant CD34 and CD43 signals were shown on the endothelial layer of wild-type and heterozygous SCL^+/Δ^ aorta accompanied by a few dispersed CD31^+^/CD34^+^/CD43^+^ cells in the nearby cavities (Figure 5A,B). Nonetheless, CD31 signals were detected in a limited manner on the endothelial layer of homozygous *SCL*^Δ/Δ^ aorta (Figure 5C).

## 3. Discussion

In contrast to the *SCL* null murine embryos where primitive hematopoiesis was blocked [15,16], primitive red blood cells were present in *SCL*^Δ/Δ^ yolk sac and embryo so as to survive to E10.5 (Figure 1B,C). Such results suggest that SCL-S may be sufficient to turn on the primitive hematopoiesis which is SCL-L independent. The expression of SCL-S could be detected on the endothelial layer of the aorta at E9.5 (Figure 3B), however, the HEC marker CD34 and HSC marker CD43 were still not detectable at the same place and time (Figure 4A). The last two markers became detectable on the endothelial layer of aorta at E10.5 (Figure 5A). That is to say, SCL-S was expressed before the maturation of aortic HECs where definitive hematopoiesis begins after E10.5. Although high levels of SCL-S proteins accumulated in the vascular endothelial cells of *SCL*^Δ/Δ^ mutants, neither CD34 nor CD43 positive HECs could be found up to E10.5 (Figure 5C). Only the SCL-L expressing cells could sprout from HECs into blood vessels and transported by blood flow. Besides SCL-C and SCL-N (Figure 3B), these dispersed cells were also stained by CD31, CD34, CD43, and c-Kit antibodies (Figure 4). Most of them were CD45 positive (Figure 4C). According to these results, it is supposed that SCL-S may be involved in the early stage of HEC development for HECs’ maturation, and SCL-L is necessary for EHT, demonstrated by HSCs budding from HECs in pro-definitive and definitive hematopoiesis. The conventional vascular endothelial marker CD31 was not detected on the endothelial layer of the wild-type and heterozygous *SCL*^+/Δ^ aorta at E10.5, instead of the *SCL*^Δ/Δ^ mutant (Figure 5C), yet it was strongly stained on the dispersed cells in the nearby cavities of the aorta (Figure 5A,B). It could be proposed that the endothelial lineage was set as the default in the endothelial layer and SCL-S acted as a repressor for the differentiation of the endothelium? The other question is why the sprouted cells expressed CD31. Some recent reports that the erythro-myeloid progenitors were CD31^+^/CD45^+^ may provide the answer [33,34].

As mentioned above, two isoforms of the SCL protein and mRNA were differentially expressed in the zebrafish [17,19] and mouse [21,22,23] embryos, temporally and spatially. The full length SCL-L protein is composed of a Gly/Pro-rich N-terminal domain, a basic-helix-loop-helix (bHLH) domain, from amino acid residue 185 to 249, and a short Gly/Pro/Ser rich C-terminus. Both the N- and C-terminal domains are structurally disordered, and the C-terminal domain has a high propensity score of liquid–liquid phase separation (LLPS) or protein phase separation (PPS) (Appendix A). The middle bHLH domain was involved in the heterodimerization with the counterpart of E47 and LMO2 as well as DNA binding to E-box elements [35,36]. The SCL-S protein is composed of the bHLH domain and the C-terminal domain. The intrinsically disordered regions (IDRs) have been deeply explored in recent years [37,38]. It is common rather than unusual for transcription factors to contain IDRs [39,40]. IDRs are involved in protein communications via weakly and dynamically transient interactions for various scenarios [41,42,43,44] to direct transcription factor in vivo binding specificity [45,46]. In order to figure out partnerships between the N-terminal and C-terminal IDRs of SCL and their associated transcription factors, the interactomes of the N-terminal and C-terminal IDRs at different developmental stages of hematopoiesis are interesting and will be explored.

## 4. Materials and Methods

### 4.1. Generation of SCL Gene Edited Mice by RNA Micro-Injection

The Cas9 mRNA and spacer-guide RNA (Sp-gRNA) expressing vectors were constructed as described previously [27]. Two oligonucleotide pairs, SclSpF1/R1 and SclSpF2/R2, corresponding to the spacers selected for the Sp-gRNA expressing vector’s construction are listed in Table 1. The Cas9 mRNA and Sp-gRNAs were mixed and adjusted to 50 to 100 ng/μL and each 10 to 30 ng/μL, respectively, before microinjection into the cytoplasmic region of the fertilized eggs. The injected eggs were transferred into a drop of M16 medium and incubated in a CO_2_ incubator. The fertilized eggs reaching the two-cell developmental stage were collected on the second day and then transfered to the oviducts near the ampulla of the surrogate ICR mothers. A live F_0_ male founder could steadily mate with wild-type FVB female mice and deliver the deleted *SCL* allele (Figure 1A) to about half of the offspring. This line was backcrossed for three rounds to reduce the possibility of off-targeting. The mice were bred and operated on at the GLP facility of the Agricultural Technology Research Institute (ATRI) under the control of the Association for Assessment and Accreditation of Laboratory Animal Care (AAALAC) rules and the animal experiments were approved by the Animal Care and Utilization Committee of ATRI.

### 4.2. Genotyping

The DNA fragment including the selected protospacers in exon 4 of the mouse *SCL* gene was amplified by PCR with forward primer SclF1 and reverse primer SclR1 (Table 1) annealing at 56 °C for 35 cycles. The sizes of the PCR products were analyzed by 1.2% agarose gel electrophoresis. The wild-type and deleted *SCL* allele were indicated by DNA bands of 623 and 505 bp, respectively.

### 4.3. Whole-Mount Immunofluorescence Staining

Mouse E9.5 embryos and yolk sacs were isolated from oviducts under a dissecting microscope (Zeiss Stemi 508, Jena, Germany) and transferred to PBS in a plate. The embryos and yolk sacs were fixed in 4% paraformaldehyde at 4 °C overnight with gentle agitation. The samples were transferred to PBS-azide buffer (0.02% sodium azide in PBS) and stored at 4 °C before use. The embryos and yolk sacs were permeabilized with PBS supplemented with 1% Triton X-100 (J. T. Backer) at room temperature three times each for 1 h and incubated in blocking solution (PBS/0.3% Triton X-100/4% normal donkey serum) at room temperature for 6 h. Then, the samples were incubated with diluted primary antibodies (rat mAb anti-CD31, BioCare Medical, CM 303; mouse mAb anti-CD34, Santa cruz, sc-74499) with 25 to 100 fold dilution in PBST (PBS/0.3% Triton X-100) at 4 °C for 24 h and washed in PBST three times for 1 h each. The samples were treated with Alexa Fluor 488- or Alexa Fluor 594-conjugated secondary antibodies (200 fold dilution) at 4 °C for 24 h with the same procedures and counterstained with 20 μg/mL DiD (1,1′-dioctadecyl-3,3,3′,3′-tetramethylindodicarbocyanine perchlorate, Invitrogen, Cat. No. D307) in PBS supplemented with 0.5% TWEEN^®^ 20 (P1379, Sigma-Aldrich, St. Louis, MO, USA) at 25 °C for 6 h. The drained samples were soaked in FocusClear [47] overnight and then transferred to the center of a ring washer of 440 μm thickness set on a slide. The space was filled with fresh FocusClear before coverslip setting. The fluorescence images were taken by a LSM780 (Carl Zeiss, Jena, Germany) confocal scanning laser microscope with tile scan plus Z-stack.

### 4.4. Embryo Section and Immunofluorescence Staining

Embryos were fixed in 4% paraformaldehyde at 4 °C overnight with gentle agitation. After dehydration, the embryos were embedded in paraffin for sectioning. Paraffin-embedded sections (5 μm thick) were de-waxed and heat-induced antigen retrieval was performed by Dako target retrieval solution in a pressure cooker. Sections were blocked with 2% BSA in PBS for 30 min. Sections were incubated with primary antibodies (Table 2) with 50 fold dilution at 4 °C overnight, washed with PBS containing 0.1% Triton X-100 for 10 min three times, then incubated with secondary antibodies (Alexa Fluor 488 goat anti-rabbit IgG or 594 goat anti-mouse IgG, 1:500) for 1 h. After washing three times, sections were stained with 0.5 μg/mL DAPI for 10 min and washed. The sections were mounted in 50% glycerol in PBS and analyzed by confocal microscopy.

## 5. Concluding Remarks

The short isoform of SCL protein (SCL-S) is sufficient to sustain primitive hematopoiesis and the full-length isoform (SCL-L) is necessary to launch definitive hematopoiesis in mouse embryos.

## Figures and Tables

**Figure 1 ijms-24-06427-f001:**
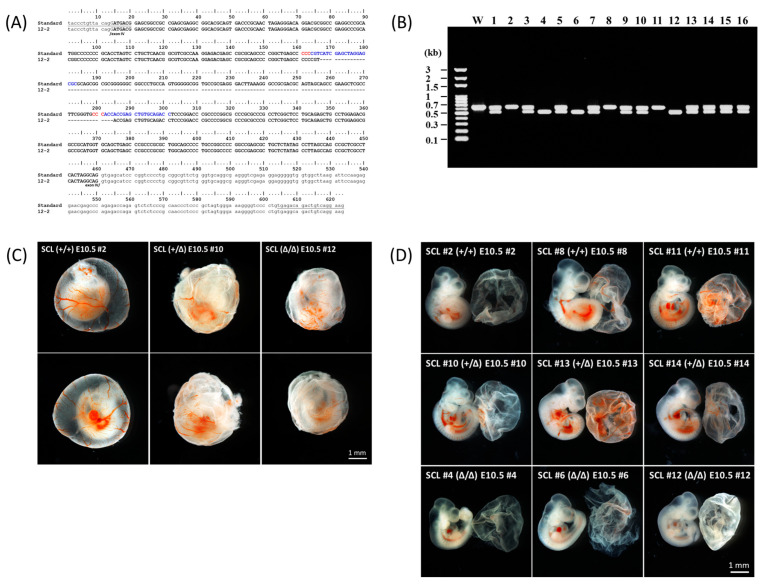
Mutation of mouse *SCL* gene with a deletion in exon IV (**A**) The sequences of the PCR amplification region are presented, in which the protospacers and PAM are shown in blue and red letters, respectively, and the 118 bp deletion found in the F_0_ founder for breeding is located by a dashed line. Primers for PCR are underlined. (**B**) A litter of embryos taken as an example. In this case, littermates 2, 8, and 11 were wild type; littermates 1, 3, 5, 7, 9, 10, 13, 14, 15, and 16 were heterozygous; and littermates 4, 6, and 12 were homozygotes. (**C**) Red blood cells and the blood vessel systems are illustrated in each two intact yolk sacs of the wild type, heterozygous *SCL*^+/Δ^, and homozygous *SCL*^Δ/Δ^ E10.5 embryos. (**D**) Blood in the vessels and heart could be clearly observed in each of the three wild type, heterozygous *SCL*^+/Δ^, and homozygous *SCL*^Δ/Δ^ E10.5 embryos.

**Figure 2 ijms-24-06427-f002:**
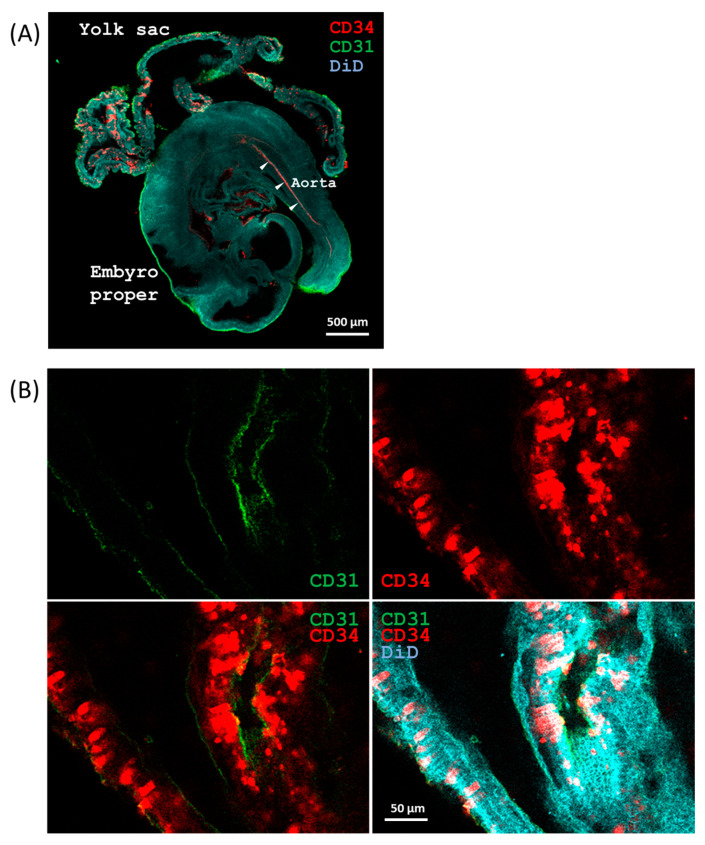
Whole mount immunofluorescence staining of an E10 murine yolk sac and embryo. (**A**) A set of E10 yolk sacs and embryo proper were whole mount stained with CD31 and CD34 antibodies to locate the vascular endothelia and HECs. The ventral side of the aorta which was revealed by the line of CD34 signal, is indicated by arrow heads. (**B**) Image of the yolk sac with higher magnification performed to unveil the rectangular and round shapes of CD34^+^ cells which were present in CD31 defined tubular structures in the yolk sac.

**Figure 3 ijms-24-06427-f003:**
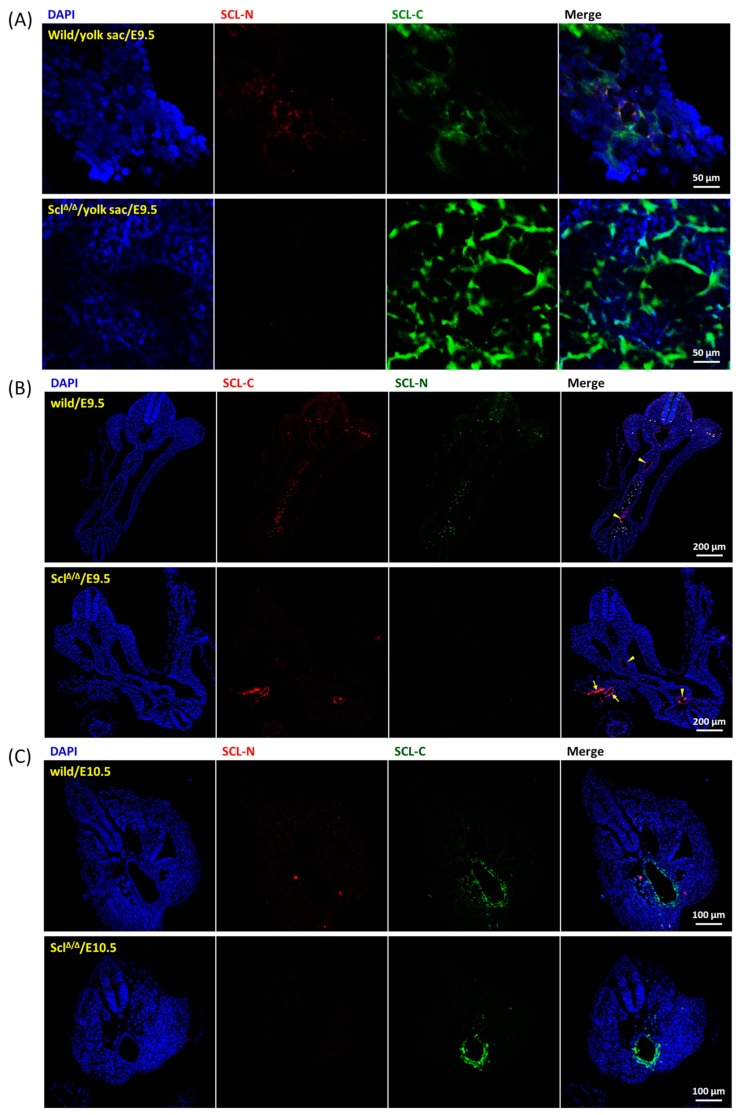
Expression patterns of SCL-C and SCL-L in wild-type and *SCL* deletion embryos. (**A**) A piece of E9.5 wild-type and *SCL*^Δ/Δ^ yolk sac were labeled with SCL-N and SCL-C antibodies to uncover the cells expressing SCL-L and/or SCL-S. A fraction of the SCL-C positive cells was labeled by SCL-N antibody in the wild-type sample (upper panels), however, only, but much stronger, SCL-C signals were detected in the *SCL*^Δ/Δ^ yolk sac (lower panels). (**B**) Transverse E9.5 embryonic sections including the aorta region were prepared and incubated with SCL-N and SCL-C antibodies. The endothelial layer of the wild-type section was stained by SCL-C alone, whilst overlapped SCL-C and SCL-N signals were observed on the dispersed cells inside the cavities of the trunk. Brighter SCL-C signals were found on the endothelial layer of the *SCL*^Δ/Δ^ aorta. The aorta regions are indicated by arrow heads and the endothelial layer of extra embryonic vessels is indicated by arrows. (**C**) Transverse E10.5 embryonic sections including the aorta region were incubated with SCL-N and SCL-C antibodies. The endothelial layer of the wild-type section was stained by SCL-C alone, whilst a few dispersed cells inside the trunk cavities were doubly stained by SCL-C and SCL-N. Brighter SCL-C signals were found on the endothelial layer of the *SCL*^Δ/Δ^ aorta.

**Figure 4 ijms-24-06427-f004:**
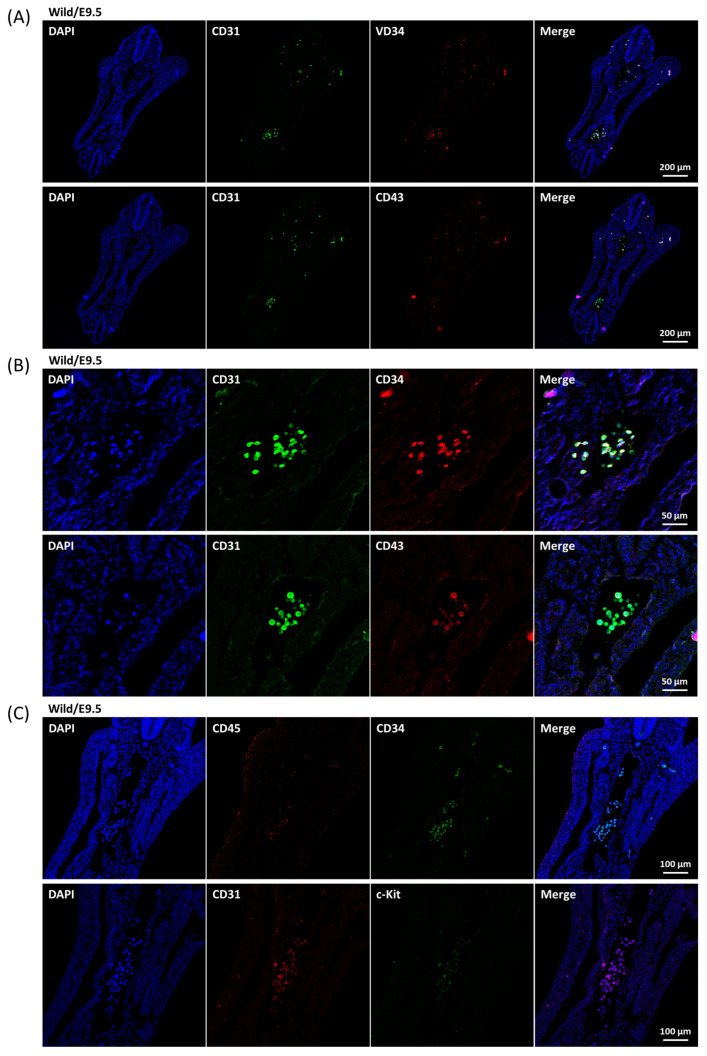
Expression patterns of CD31, CD34, and HSC markers in E9.5 wild-type embryo. (**A**) Besides the SCL-C and SCL-N, the wild-type transverse E9.5 embryonic sections were prepared and co-incubated with either CD31 and CD34 (upper panels) or CD31 and CD43 (lower panels) antibodies. Only the dispersed cells were stained. (**B**) Image at higher magnification focusing on the major groups of the dispersed cells. (**C**) The wild-type transverse E9.5 embryonic sections were co-incubated with either CD34 and CD45 (upper panels) or CD31 and c-Kit (lower panels) antibodies.

**Figure 5 ijms-24-06427-f005:**
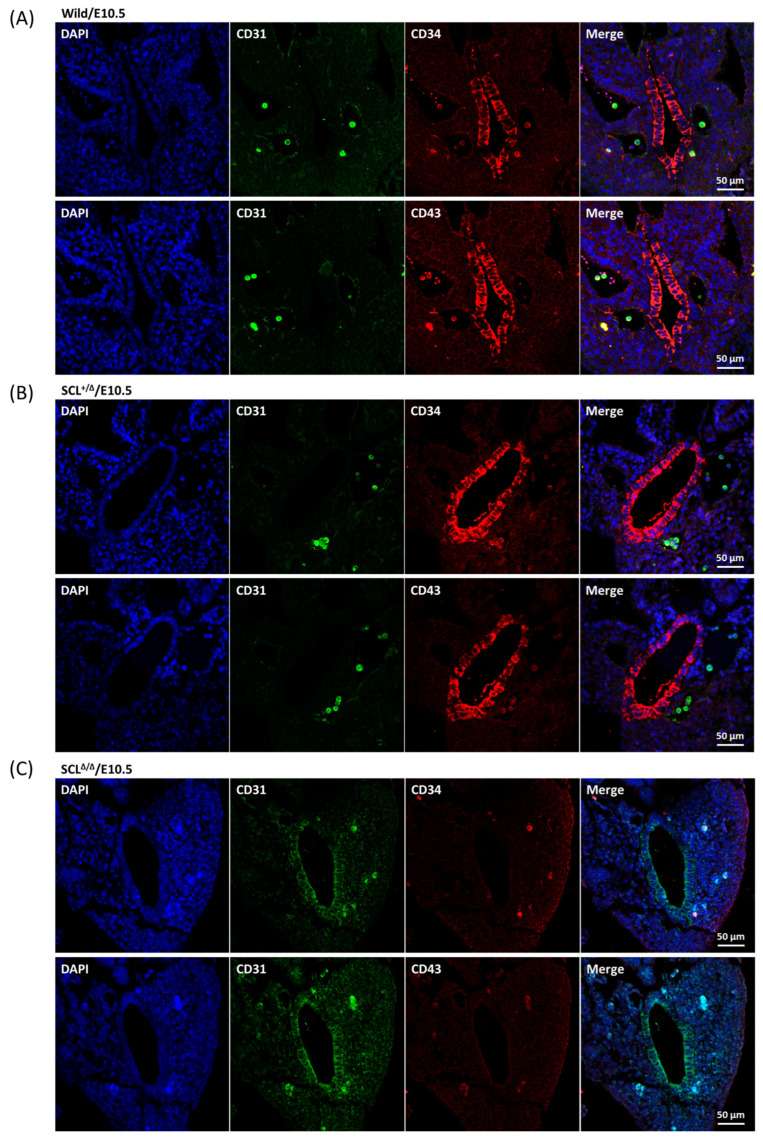
Expression patterns of CD31 and CD34 in E10.5 wild-type and *SCL* deletion embryos. Transverse sections of E10.5 wild-type (**A**) and *SCL*^+/Δ^ (**B**), and *SCL*^Δ/Δ^ (**C**) embryo proper were co-stained with either CD31 and CD34 (upper panels) or CD31 and CD43 (lower panels). A layer of the CD31^−^/CD34^+^ and/or CD31^−^/CD43^+^ cells emerged on the endothelial layer of the aorta of the wild-type and the heterologous Scl^+/Δ^ E10.5 embryos, meanwhile several CD31^+^/CD34^+^ and/or CD31^+^/CD43^+^ cells are observed in the other lumens of these sections (**A**,**B**). On the other hand, a layer of CD31^+^ cells without CD34 or CD43 signals was shown in the E10.5 *SCL*^Δ/Δ^ sections (**C**). Regardless of the genotypes, a few CD31^+^/CD34^+^ and/or CD31^+^/CD43^+^ cells were stained in the other cavities beyond the aorta (**A**–**C**).

**Table 1 ijms-24-06427-t001:** Oligonucleotides used in this study.

Primer Name	DNA Sequence
SclF1	TACCCTGTTACAGGATGACG
SclR1	CTTCCTGACAGTCTGTCTCA
SclSpF1	CGTC GCGCTCCTAGCTCGATGACG GTTTTAGAGCTAGAAAT
SclSpR1	TGCTATTTCTAGCTCTAAAAC CGTCATCGAGCTAGGAGCGC
SclSpF2	CGTC GGTCTGCACAGCTCGGTGGT GTTTTAGAGCTAGAAAT
SclSpR2	TGCTATTTCTAGCTCTAAAAC ACCACCGAGCTGTGCAGACC

**Table 2 ijms-24-06427-t002:** Primary antibodies used in this study.

Antibodies	Species/Isotype	Brand	Cat. No.
Anti-Scl (N-terminus)	Rabbit polyclonal antibody	LSBio	LS-B14706
Anti-Scl (C-terminus)	Mouse monoclonal antibody (IgG1, κ)	Santa Cruz	sc-393287
CD31	Rat monoclonal antibody (IgG2a, κ)	BioCare Medical	CM 303
CD31	Rabbit polyclonal antibody	GeneTex	GTX130274
CD34	Mouse monoclonal antibody (IgG1, κ)	Santa Cruz	sc-74499
CD43	Mouse monoclonal antibody (IgG1, κ) [DF-T1]	GeneTex	GTX73633
c-Kit	Rat monoclonal antibody (IgG2b)	Santa Cruz	sc-19619
CD45 (H-230)	Rabbit polyclonal antibody	Santa Cruz	sc-25590

## Data Availability

Not applicable.

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
