# Peer review of "The Role of SCL Isoforms in Embryonic Hematopoiesis"

_ijms, 2023, doi:10.3390/ijms24076427_

Round 1

Reviewer 1 Report

The authors report design and establishment of a KO mouse which has deleted the large isoform of the hematopoietic transcription factor TAL1. The topic is interesting and the work could give corresponding answers.

However, the manuscript has major deficiencies which require fundamental revision. 

1. The text (except the introduction) is confusing and shows no clear route to follow. Most of the parts belong to the material and methods section. The reader is lost in details. Possibly this manuscript belongs to a journal which is focussed on methods. 

2. The literature is not indicated in brackets, making the text difficult to read. The figure legends (Fig 2 and Fig 3) are incomplete. 

3. The authors do not show expression data (RNA and protein !!!) of TAL1, both in the according tissues from wild type animals and knockout mice. They just refer to the reported situations in cell lines and fish. That is essential for this kind of study. 

Author Response

We would like to thank the reviewers’ constructive suggestions, and Please see the attachment with the point-by-point response.

Reviewer 2 Report

In this paper, Chuang et al generated the mutant mice lacking long isoforms of SCL/tal1 protein. The author showed that long isoforms of SCL/tal1 are dispensable for the generation of primitive erythrocytes, but are required for definitive hematopoiesis. This study is done very simply and the data are clearly shown. However,  a couple of further experiments are recommended to support the authors’ conclusion. Specific comments are provided below.

Major points:

1) Expression of CD41 or Runx1 should be examined for the aortic region. These are more specific markers for definitive hematopoiesis in E10.5 embryo proper.

2) There are “transient” definitive hematopoietic progenitors found in E9.5 yolk sac. This reviewer would like to know whether transient definitive progenitors like EMPs and MPPs (lympho-myeloid progenitors) are present or not in the E9.5 KO yolk sac. The authors can examine CD45 expression to monitor the presence of transient definitive hematopoiesis in this stage of yolk sac tissue.

Minor point:

1) There are no quantitative methods provided to examine the primitive erythropoiesis. The authors may add an assay to precisely measure the quantity of primitive erythrocytes in the KO yolk sac  (eg, flow-cytometry, globin expressions). 

Author Response

(The authors gave the same response as above.)

Round 2

Reviewer 1 Report

The manuscript is still very hard to read. The text contains too much experimental details. There are several parts in the Results and Discussion sections which have to be transfered into the Methods-section. The text itself still requires revisions. 

Author Response

(The authors gave the same response as above.)

Reviewer 2 Report

there are no specific comments at this stage

Author Response

(The authors gave the same response as above.)

Round 3

Reviewer 1 Report

The authors have improved the text of their manuscript. However, I still have some points which require correction:

1. In the introduction you have described in detail TAL1 in fish. However, please expand rather the description about TAL1 and its isoforms in mouse.

2. The study reports about knockout of TAL1. It is important to confirm this knockout at the DNA, RNA and protein level. Please show these data in figures.

3. In the discussion, lines 215-220 repeat what is written in the introduction. Skip this repetition or add additional informations at this position.

Author Response

We would like to thank the reviewers’ constructive suggestions, and Please see the attachment with the point-by-point response.

1. In the introduction you have described in detail TAL1 in fish. However, please expand rather the description about TAL1 and its isoforms in mouse.

Response: Although the development of embryonic hematopoiesis is quite conserved in vertebrates, only limited clues of mammalian SCL isoforms have been introduced hitherto. We rewrite lines 62-72 to expand the description.

2. The study reports about knockout of TAL1. It is important to confirm this knockout at the DNA, RNA and protein level. Please show these data in figures.

Response:  We show the DNA data of the TAL1 knockout genotyping in Figure 1B, and add lines 94-95, 103-105, 250-253, and 263-264 to provide an explanation. The data on protein levels have been presented in Figure 3, as previously mentioned.

3. In the discussion, lines 215-220 repeat what is written in the introduction. Skip this repetition or add additional informations at this position.

Response:  We skip the redundancy part and rewrite it as lines 223-225 in the Discussion.
